# The genetic architecture of the human thalamus and its overlap with ten common brain disorders

Torbjørn Elvsåshagen [1,2,3✉], Alexey Shadrin [1,3], Oleksandr Frei[1,3,4], Dennis van der Meer[1,5], Shahram Bahrami[1,3], Vinod Jangir Kumar[6], Olav Smeland [1,3], Lars T. Westlye [1,7,8], Ole A. Andreassen [1,3,8] & Tobias Kaufmann [1,3,9✉]

The thalamus is a vital communication hub in the center of the brain and consists of distinct nuclei critical for consciousness and higher-order cortical functions. Structural and functional thalamic alterations are involved in the pathogenesis of common brain disorders, yet the genetic architecture of the thalamus remains largely unknown. Here, using brain scans and genotype data from 30,114 individuals, we identify 55 lead single nucleotide polymorphisms (SNPs) within 42 genetic loci and 391 genes associated with volumes of the thalamus and its nuclei. In an independent validation sample ($n = 5173$) 53 out of the 55 lead SNPs of the discovery sample show the same effect direction (sign test, $P = 8.6e\text{-}14$). We map the genetic relationship between thalamic nuclei and 180 cerebral cortical areas and find over-lapping genetic architectures consistent with thalamocortical connectivity. Pleiotropy analyses between thalamic volumes and ten psychiatric and neurological disorders reveal shared variants for all disorders. Together, these analyses identify genetic loci linked to thalamic nuclei and substantiate the emerging view of the thalamus having central roles in cortical functioning and common brain disorders.

[1] NORMENT, Division of Mental Health and Addiction, Oslo University Hospital, Oslo, Norway. [2] Department of Neurology, Division of Clinical Neuroscience, Oslo University Hospital, Oslo, Norway. [3] Institute of Clinical Medicine, University of Oslo, Oslo, Norway. [4] Center for Bioinformatics, Department of Informatics, University of Oslo, Oslo, Norway. [5] Faculty of Health, Medicine and Life Sciences, School of Mental Health and Neuroscience, Maastricht University, Maastricht, The Netherlands. [6] Max Planck Institute for Biological Cybernetics, Tübingen, Germany. [7] Department of Psychology, University of Oslo, Oslo, Norway. [8] K.G. Jebsen Center for Neurodevelopmental Disorders, University of Oslo, Oslo, Norway. [9] Tübingen Center for Mental Health, Department of Psychiatry and Psychotherapy, University of Tübingen, Tübingen, Germany. ✉email: torbjorn.elvsashagen@medisin.uio.no; tobias.kaufmann@medisin.uio.no

Recent studies indicated that the thalamus has a broader role in cognition than previously assumed[1–5]. Cognitive neuroscience is therefore shifting focus to how the thalamus regulates cortical activity and supports higher-order functions such as working memory[6], attentional control[7], and visual processing[8]. There is also a growing appreciation of thalamic contributions to the pathogenesis of neurological and psychiatric disorders, including dementias[9], Parkinson's disease (PD)[10], schizophrenia (SCZ)[11], and bipolar disorder (BD)[12]. Despite the importance for human cognition and disease, the genetic architecture of the thalamus and its relationship with cortical structure and common brain disorders remain largely unknown.

The thalamus can be divided into nuclei that mainly relay peripheral information to the cerebral cortex and higher-order nuclei that modulate cortical functions[1,4]. Two recent studies found one[13] and two[14] genetic loci associated with whole thalamus volume, yet there is no genome-wide association study (GWAS) of thalamic nuclei. In this work, we further identify genetic loci and candidate genes for volumes of the whole thalamus and its nuclei. We detect overlap between genetic architectures of the thalamic nuclei and 180 cortical areas consistent with thalamocortical structural connectivity. Moreover, pleiotropy analyses between thalamic volumes and ten neurological and psychiatric disorders reveal shared variants for all disorders. Together, these analyses provide insights into the genetic underpinnings of human thalamic nuclei and substantiate the emerging view of the thalamus having central roles in common brain disorders.

## Results and discussion

We accessed brain magnetic resonance imaging (MRI) data from $n = 30,432$ genotyped White British from the UK Biobank[15]. The MRI data were segmented into the whole thalamus and six thalamic nuclei groups—anterior, lateral, ventral, intralaminar, medial, and posterior—using Bayesian thalamus segmentation[16,17] (Fig. 1a). We removed data sets with segmentation errors and insufficient data quality ($n = 318$) after visually inspecting the segmentations for each of the 30,432 participants. The resulting 30,114 data sets comprised the discovery sample (52% females; age range 45–82 years).

We conducted GWAS with PLINK[18] on whole thalamus and the six nuclei volumes accounting for age, age-orthogonalized age-squared, sex, scanning site, intracranial volume, and the first 20 genetic principal components. The thalamic nuclei GWAS also accounted for whole thalamus volume, thus revealing genetic signals beyond commonality in volume[19,20]. The thalamic nuclei GWAS were also run without covarying for whole thalamus volume and are presented in the Supplementary Information.

Single-nucleotide polymorphism (SNP)-based heritability estimated using linkage disequilibrium (LD) score regression[21] was 25% for whole thalamus and 18–32% for the six nuclei groups (Fig. 1b). We found genome-wide significant hits for all seven volumes (Bonferroni-corrected $P < 5e − 8/7 = 7e − 9$) and identified a total of 55 lead SNPs in 50 genomic loci (Fig. 1c, d and Supplementary Data 1). Seven loci were associated with whole thalamus; 3, 11, and 5 loci with anterior, lateral, and ventral nuclei; and 12, 6, and 6 loci were associated with intralaminar, medial,

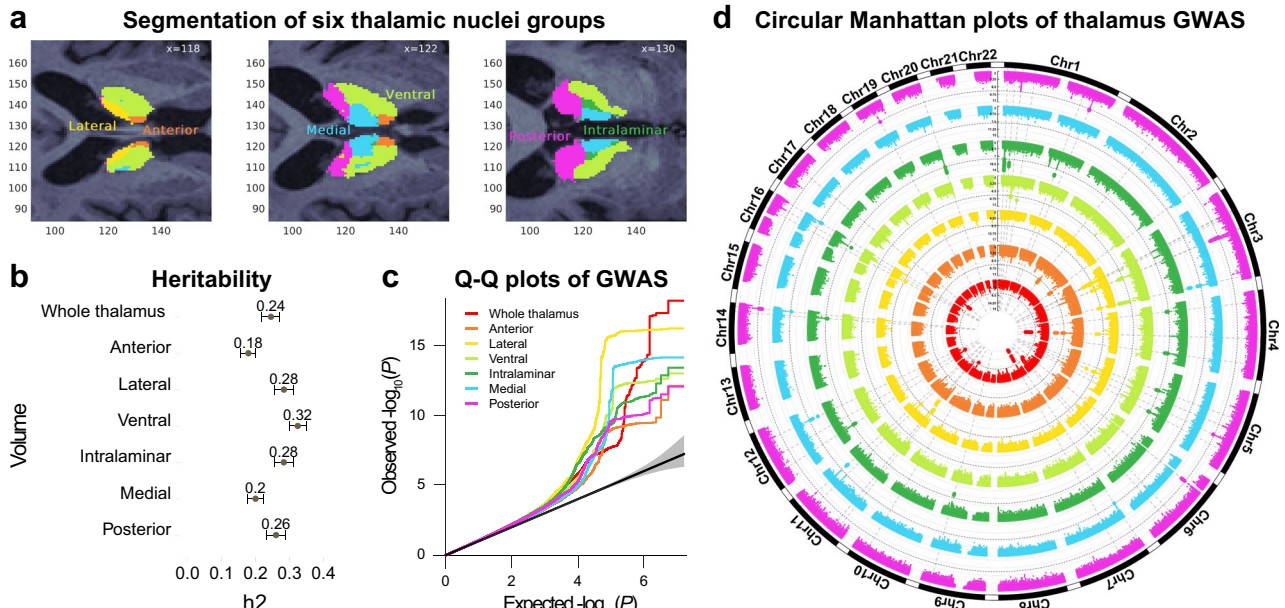

**Fig. 1 Analysis of the GWAS discovery sample identifies 42 loci associated with thalamic volumes. a** The thalamus was segmented into six nuclei groups—anterior, lateral, ventral, intralaminar, medial, and posterior nuclei—using Bayesian thalamus segmentation[16, 17]. All data sets were assessed by visually inspecting axial view figures of the segmentations for each participant and we removed sets with segmentation errors, insufficient data quality, and pathologies. The nuclei volumes in the left and right thalamus were summed and these were used in the analyses. **b** Heritability estimates for the thalamic volumes in the discovery sample of $n = 30,114$ participants from the UK Biobank. All thalamic volumes showed substantial heritability. Data are presented as mean ± SE. **c** Q–Q plots for the thalamic volumes of the discovery sample. **d** Circular Manhattan plots of GWAS for thalamus volumes of the discovery sample. The innermost plot reflects the GWAS of whole thalamus volume, whereas from center to the periphery, the plots indicate the GWAS of the anterior, lateral, ventral, intralaminar, medial, and posterior nuclei, respectively. Black circular dashed lines indicate genome-wide significance (two-sided $P < 7e − 9$). Horizontal Manhattan plots for the seven volumes are shown in Supplementary Fig. 2. The colors in **a**, **c**, and **d** indicate the same volumes, i.e., red color reflects whole thalamus; orange, yellow, and light green indicate the anterior, lateral, and ventral nuclei, respectively; whereas dark green, blue, and magenta reflect intralaminar, medial, and posterior nuclei volumes, respectively. GWAS, genome-wide association studies; h2, heritability.

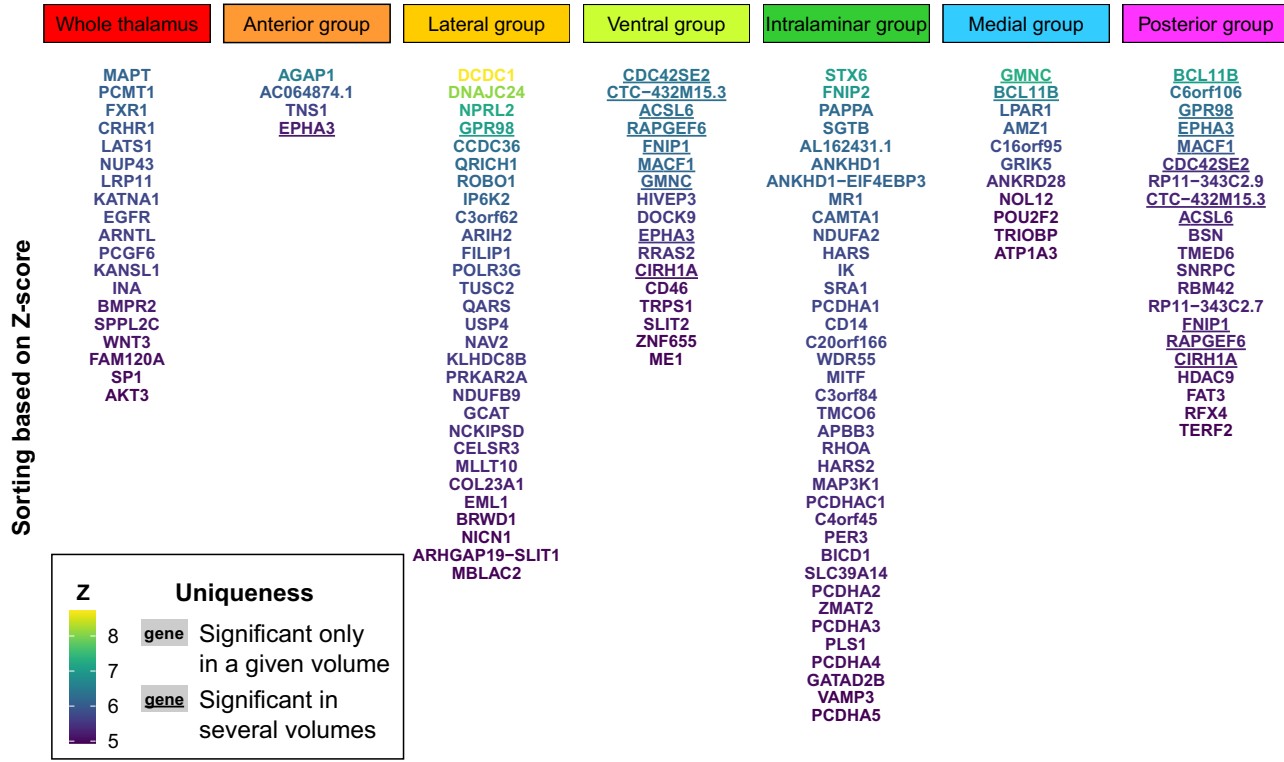

**Fig. 2 GWGAS identifies 127 unique genes associated with thalamic volumes.** Nineteen genes were associated with whole thalamus; 4, 29, and 17 genes were associated with volumes of the anterior, lateral, and ventral nuclei; and 37, 11, and 21 genes were associated with intralaminar, medial, and posterior nuclei volumes, respectively. Lighter font color and higher position for gene names indicate greater Z-score. Underlined gene names designate genes that were significantly associated with more than one volume, whereas gene names not underlined indicate genes associated with only one volume. Additional results of the GWGAS are presented in Supplementary Data 12. GWGAS, genome-wide gene-based association analysis.

and posterior nuclei volumes, respectively. Eight loci were associated with more than 1 volume, resulting in 42 unique thalamus-associated loci. One of these (rs76928645 on chromosome 7) was associated with whole thalamus volume in a recent study[13]. Notably, 37 of the 42 genetic loci were associated with only one volume. The genomic inflation factors for the seven GWAS are listed in Supplementary Data 2. The strongest genomic inflation was observed for the ventral nuclei ($\lambda = 1.168$), whereas the anterior nuclei showed the weakest inflation ($\lambda = 1.105$).

Three lead SNPs had a combined annotation-dependent depletion (CADD) score > 15, which indicates deleterious protein effects[22]. rs13107325 (score = 23.1) was associated with anterior and lateral nuclei volumes, is located within the metal ions transporter gene *SLC39A8* on chromosome 4, and is linked to cognitive performance, PD, and SCZ[23,24]. The gene nearest to rs12146713 on chromosome 12 (associated with medial nuclei volume; score = 19.6) is *NUAK1*, which regulates the Tau protein level[25]. Cerebral Tau accumulation is a defining characteristic of Alzheimer's disease (AD) and other neurodegenerative disorders[26]. rs951366 on chromosome 1 was significant for posterior nuclei volume (score = 15.7) and is associated with PD[27]. Further GWAS results are provided in Supplementary Figs. 1–4 and Supplementary Data 3–9.

For GWAS replication, we used MRI and SNP data from an additional 5173 White British (51% females; age range 46–81 years) from the UK Biobank[15]. We found that 53 out of the 55 lead SNPs from the discovery GWAS had the same effect direction in the replication (sign test; $P = 8.6e − 14$). Moreover, 32 of the discovery lead SNPs had uncorrected $P < 0.05$, whereas 23 had uncorrected $P > 0.05$, in the replication (Supplementary Data 10).

To assess the tissue and cell-type specificity of the thalamic GWAS findings, we used LD-score regression applied to specifically expressed genes (LDSC-SEGs)[28]. We found that volumes of the whole thalamus, and intralaminar and medial nuclei groups were enriched for central nervous system (CNS) tissues and cell types, yet none of these associations remained significant after Bonferroni correction (Supplementary Fig. 5).

To further examine the biological significance of the GWAS results, we used positional, expression quantitative trait loci (eQTL), and chromatin interaction mapping in Functional Mapping and Annotation of GWAS (FUMA)[29] to map candidate SNPs to genes. This identified 336 unique genes across the 7 volumes (Supplementary Data 11 and Supplementary Fig. 6). We conducted genome-wide gene-based association analyses (GWGAS) using MAGMA[30] and detected 127 unique genes across the thalamic volumes (Fig. 2, Supplementary Fig. 7, and Supplementary Data 12). The GWGAS gene most strongly associated with whole thalamus volume was *MAPT*. *MAPT* codes for Tau protein in neurons, is implicated in the pathogenesis of neurodegenerative disorders[26], and is linked to general cognitive ability[31]. The most strongly associated GWGAS gene for the thalamic nuclei was *DCDC1*, which was linked to lateral nuclei volume. *DCDC1* is a member of the doublecortin gene family with high expression levels in the fetal brain[32], yet its functions remain largely unknown.

Fifty-five of the GWGAS genes were not mapped by the GWAS analyses, resulting in a total of 391 thalamus-linked genes. Notably, 95% of these were associated with only one volume. We conducted gene-set analyses using MAGMA[30] and found significant Gene Ontology sets for the intralaminar nuclei, implicating cell–cell adhesion, and for the lateral nuclei, involving cell growth and synapse organization (Supplementary Data 13).

We then performed protein–protein interaction analyses to explore the functional relationships between proteins encoded by

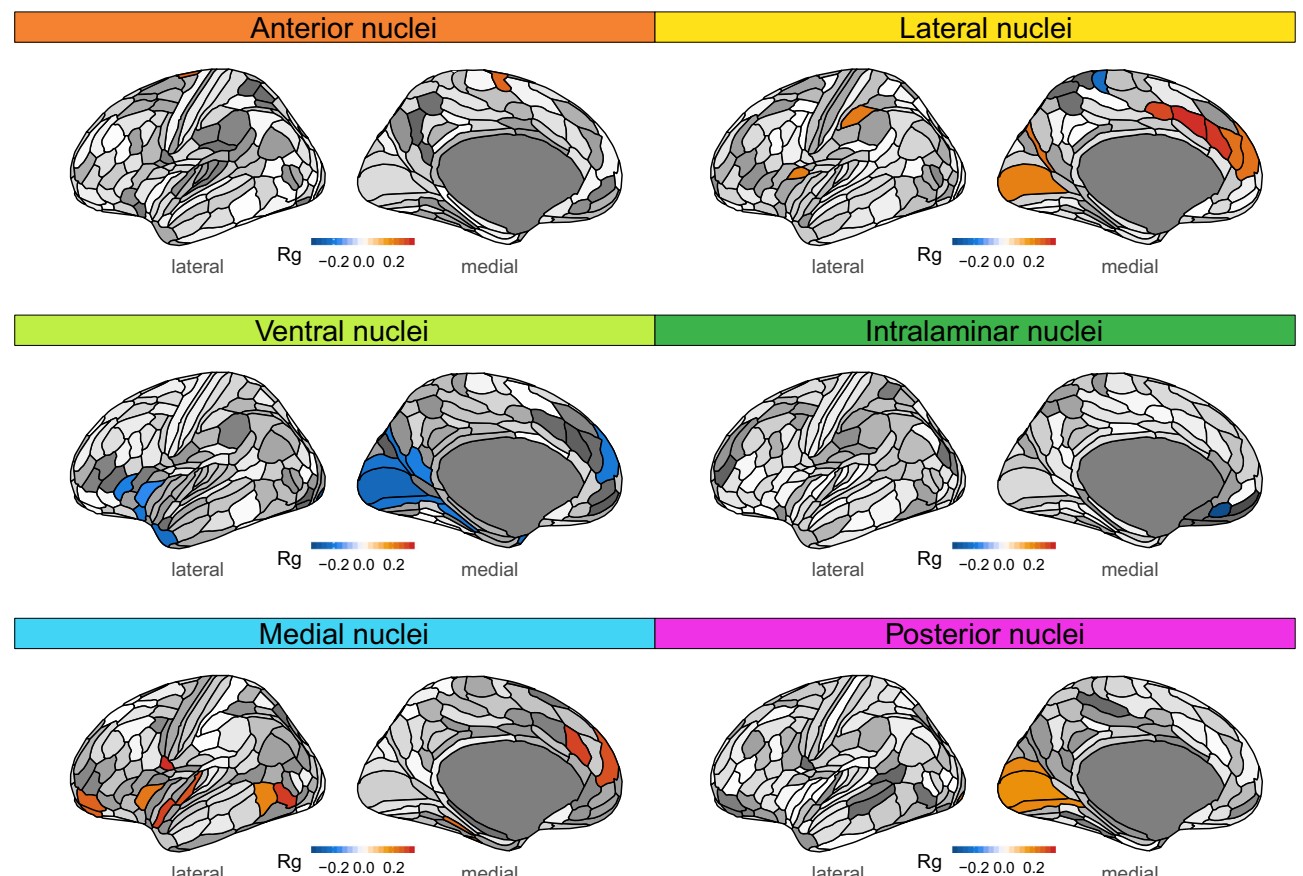

**Fig. 3 Thalamocortical genetic relationships.** We found significant genetic correlations between all thalamic nuclei and distinct cortical regions. There were significant positive genetic correlations between anterior nuclei and medial premotor cortex; between lateral nuclei and mainly medial prefrontal, anterior cingulate, parietal, and visual cortices; between medial nuclei and prefrontal and temporal cortices; and between posterior nuclei and visual cortices. We also found significant negative genetic associations between ventral nuclei and visual, prefrontal, and temporal cortical regions, and between intralaminar nuclei and rostral medial prefrontal cortex. Warm and cool colors in cortical regions indicate significant positive and negative genetic correlations, respectively, after adjusting for analyses of 180 cortical regions and 6 volumes (two-sided *P* < FDR). Corresponding statistics are provided in Supplementary Data 15. FDR, false discovery rate; Rg, genetic correlation.

the 391 candidate genes and detected a network with significantly more interactions than expected by chance (protein–protein interaction enrichment: *P* < 1e − 16; Supplementary Data 14 and Supplementary Fig. 8). The most central network nodes were EGFR, RHOA, KANSL1, and NFKB1.

The above analyses identified genetic loci linked to higher-order thalamic nuclei, which project to distinct cortical regions and support cognition[1,4,33]. For example, medial thalamic nuclei are densely interconnected with prefrontal and temporal cortices, and regulate working memory and attentional control[6,7], whereas posterior thalamic nuclei project to occipital cortices and support visual processing[8]. Recent studies suggest that well-connected brain regions exhibit stronger genetic correlations than less-connected regions[34,35], yet whether this principle applies to thalamocortical connectivity remains unknown. Thus, we ran GWAS in the discovery sample for volumes for each of the 180 cortical regions defined recently[36] and examined genetic correlations with the six thalamic nuclei volumes. These analyses revealed significant associations between specific cortical regions and each of the thalamic nuclei (Fig. 3 and Supplementary Data 15). Interestingly, we found positive correlations mainly for higher-order thalamic nuclei with cortical distributions consistent with established thalamocortical projection patterns, i.e., medial nuclei correlated with prefrontal and temporal cortices, and posterior nuclei with the visual cortex. We also found significant positive associations between the higher-order lateral nuclei[17,34] and cortical regions, consistent with their connections with medial prefrontal, anterior cingulate, parietal, and visual cortices[37,38].

Thalamic alterations are reported in a growing number of psychiatric and neurological disorders[9–12,39,40], yet the genetic relationships between the thalamus and disorders have not been clarified. We used GWAS summary statistics for attention-deficit/hyperactivity disorder (ADHD)[41], autism spectrum disorder (ASD)[42], BD[43], major depression (MD)[44,45], SCZ[46], AD[47], multiple sclerosis (MS)[48], PD[27,49], and generalized and focal epilepsy (GEP/FEP)[50], and detected significant genetic correlations between volumes and PD, BD, and MS (Fig. 4).

To further examine the polygenic overlap between thalamic volumes and the ten disorders, we performed conjunctional false discovery rate (FDR) analyses, which enable detection of genetic loci shared between traits[51–54]. Notably, we identified jointly associated loci across volumes and disorders (Fig. 5 and Supplementary Data 16), and found the largest number of overlapping loci for SCZ (66), PD (26), and BD (15), when applying a conjunctional FDR threshold of 0.05. ASD, ADHD, MD, MS, GEP, FEP, and MS had 8, 8, 17, 10, 14, 4, and 14 loci jointly associated with thalamic volumes, respectively. When using a more stringent conjunctional FDR threshold of 0.01, there were jointly associated loci for all disorders, except GEP and FEP (Supplementary Data 16).

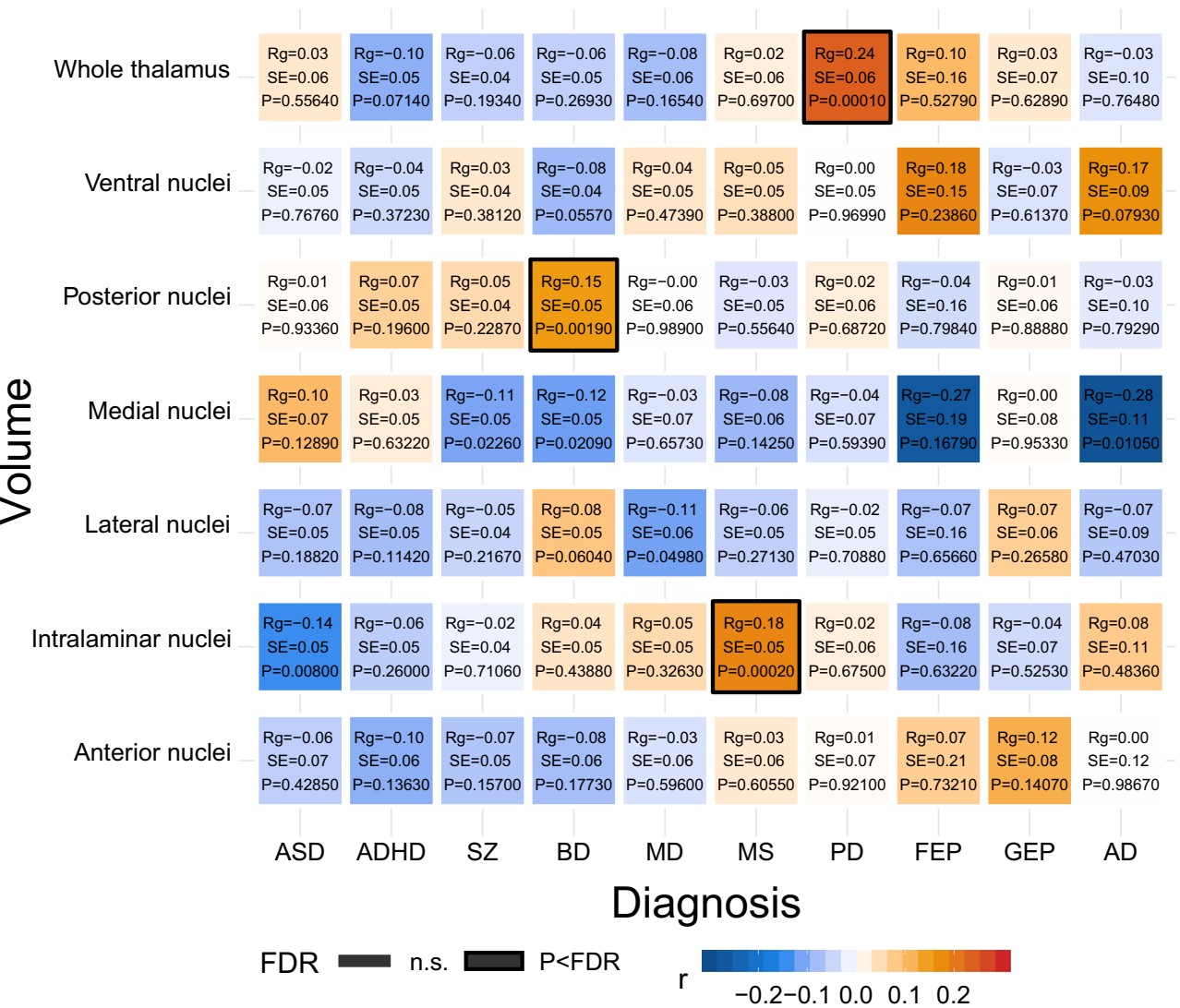

**Fig. 4 LD-score regression-based genetic correlations between thalamic volumes and ten brain disorders.** We assessed genetic correlations between thalamic volumes and ten brain disorders using LD-score regression. Warm and cool colors indicate positive and negative genetic associations, respectively. There were significant positive correlations between the whole thalamus and PD, between posterior nuclei and BD, and between intralaminar nuclei and MS, as indicated by a black frame, after FDR correcting across all 70 analyses (7 volumes x 10 disorders; two-sided *P* < FDR). AD; Alzheimer's disease. ADHD; attention-deficit hyperactivity disorder. ASD, autism spectrum disorder; BD, bipolar disorder; FDR, false discovery rate; FEP, focal epilepsy; GEP, generalized epilepsy; MD, major depression; MS, multiple sclerosis; PD, Parkinson's disease; SCZ, schizophrenia.

Further studies of how the overlapping genetic regions influence thalamus structure and disorder risk are warranted, yet several of the shared loci are noteworthy. rs2693698 was significant for SCZ and BD, and jointly associated with medial and posterior nuclei volumes. The gene nearest to rs2693698 is *BCL11B* (significant for the latter volumes also in the GWGAS) and encodes a transcription factor expressed in the fetal and adult brain[55]. *BCL11B* is thought to regulate neuron development and its mutations are associated with intellectual disability and neuropsychiatric disorders[55,56]. rs5011432 was associated with medial nuclei volume in MD and AD, and the nearest gene is *TMEM106B*, which regulates lysosome functions[57]. rs13107325, within *SLC39A8* as discussed above, was significant for both SCZ and PD, and associated with anterior, lateral, and intralaminar nuclei volumes.

In summary, our study provides insights into the genetic underpinnings of the human thalamus and identifies genetic loci linked to thalamic nuclei. We found that the majority of thalamus-linked loci and genes were associated with only one of the seven volumes, which may suggest at least partly independent genetic architectures. This demonstrates the importance of targeting individual nuclei rather than studying the thalamus as a whole, in line with similar benefits observed for other brain structures[19,20]. In addition, we found genetic correlations between the thalamic nuclei and distinct cortical regions. Notably, the positive associations were mainly found for higher-order thalamic nuclei and the cortical distributions are consistent with thalamocortical connectivity. These results are in line with emerging findings, suggesting that functionally connected brain regions exhibit stronger genetic correlations than less-connected regions[35,36]. Finally, we identified genetic loci shared between thalamus volumes and ten neurological and psychiatric disorders. Further mechanistic studies are required to clarify how the thalamus contributes to the pathogenesis of brain disorders and the pleiotropic loci identified by our analyses could inform such experiments.

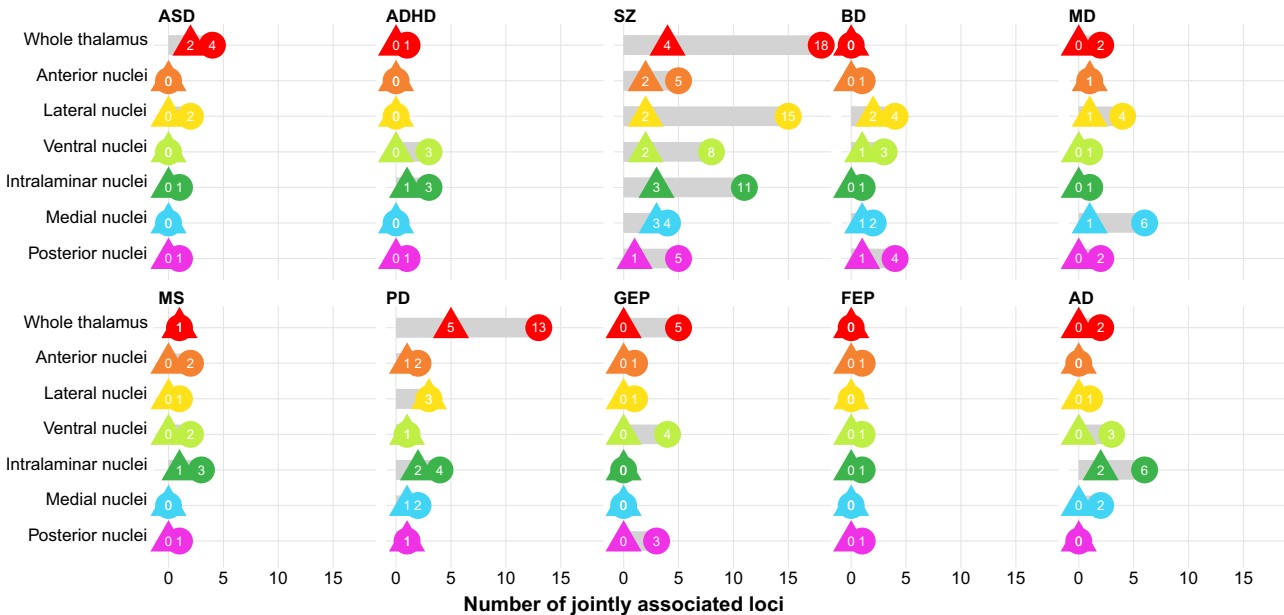

**Fig. 5 Genetic loci shared between thalamic volumes and ten brain disorders.** Conjunctional FDR analysis detected shared genetic loci across thalamic volumes and the ten disorders. The figure shows results with FDR thresholds of both 0.05 (circles) and 0.01 (triangles). We found the largest number of overlapping loci for SCZ (66), PD (26), and BD (15), when applying a conjunctional FDR threshold of 0.05. For ASD, ADHD, MD, MS, GEP, FEP, and MS, there were 8, 8, 17, 10, 14, 4, and 14 genetic loci jointly associated with thalamic volumes and disorders when using a threshold of 0.05, respectively. When using a conjunctional FDR threshold of 0.01, there were overlapping loci associated with thalamic volumes and SCZ (17), PD (14), BD (5), ASD (2), ADHD (1), MDD (3), MS (2), and AD (2), and no shared locus for GEP or FEP. AD, Alzheimer's disease; ADHD, attention-deficit hyperactivity disorder; ASD, autism spectrum disorder; BD, bipolar disorder; FDR, false discovery rate; FEP, focal epilepsy; GEP, generalized epilepsy; MD, major depression; MS, multiple sclerosis; PD, Parkinson's disease; SCZ, schizophrenia.

## Methods

**Samples, thalamus segmentations, and quality control procedures**. We included raw T1-weighted three-dimensional brain magnetic MRI data from $n = 30,432$ genotyped White British from the UK Biobank[15] in the GWAS discovery sample. The results from the discovery GWAS were also used in the analyses of genetic overlap with brain disorders and the discovery sample therefore excluded individuals with CNS diagnoses. Additional data became available to us after we had performed the main discovery analysis and we combined this data with the previously excluded data of individuals with CNS diagnoses, to maximize the sample size for a replication sample. The replication sample therefore comprised MRI and SNP data from an additional 5266 White British of the general population in the UK Biobank[15], irrespective of diagnoses. The National Health Service National Research Ethics Service (ref. 11/112 NW/0382) has approved the UK Biobank and all participants gave signed informed consent before study inclusion. The projects from which summary statistics of GWAS were used for genetic overlap analysis were each approved by the local ethics committees and informed consent was obtained from all participants[27,41–47,49,50]. The Norwegian Regional Committees for Medical and Health Research Ethics (REC South East) evaluated our pipelines that use summary statistics from published works for genetic analysis as performed in the current study and found that no additional institutional approval is needed.

The MRI data for all individuals was stored and analyzed locally at the University of Oslo. Using Bayesian thalamus segmentation based on ex-vivo MRI and histology in Freesurfer 6.0[16,17], we segmented the MRI data into the whole thalamus and six thalamic nuclei groups, i.e., anterior, lateral, ventral, intralaminar, medial, and posterior nuclei groups. These groups, as defined by Iglesias et al.[17], include the following thalamic nuclei: anterior group—the anteroventral nucleus; lateral group—the laterodorsal and lateral posterior nucleus; ventral group—the ventral anterior, ventral anterior magnocellular, ventral lateral anterior, ventral lateral posterior, ventral posterolateral, and ventromedial nucleus; intralaminar group—the central medial, central lateral, paracentral, centromedian, and parafascicular nucleus; medial group—the paratenial, reuniens, mediodorsal medial magnocellular, and mediodorsal lateral parvocellular nucleus; and the posterior group—the lateral geniculate, medial geniculate, limitans, pulvinar anterior, pulvinar medial, pulvinar lateral, and pulvinar inferior. The segmentations of these thalamic nuclei groups have high test–retest reliability, show agreement with histological studies of thalamic nuclei, and are robust to differences in MRI contrast[17]. For each nuclei group, we summed the volumes in the left and right thalamus, and these were used in all analyses.

We then manually assessed the quality and delineations of the discovery and replication sample MRI data sets by visually inspecting axial view figures of the segmentations for each participant. This procedure excluded 318 data sets from the discovery sample (due to tumors and other lesions (6%), cysts (12%), ventricle abnormalities (18%), segmentation errors (14%), and insufficient data quality (50%)) and 93 data sets from the replication sample (due to tumors and other lesions (4%), cysts (13%), ventricle abnormalities (13%), segmentation errors (9%), and insufficient data quality (61%)). Thus, the final sizes of the discovery and replication GWAS samples were $n = 30,114$ and $n = 5173$, respectively.

**GWAS studies for thalamic volumes and identification of genomic loci**. We performed GWAS on the thalamus volumes and genotype data from the participants in the GWAS discovery and replication samples, and due to data availability restricted the analyses to White British individuals. We used standard quality control procedures to the UK Biobank v3 imputed genetic data and removed SNPs with an imputation quality score < 0.5, a minor allele frequency < 0.05, missing in more than 5% of individuals, and failing the Hardy–Weinberg equilibrium tests at a $P < 1e − 6$.

GWAS was conducted for the seven thalamic volumes, i.e., volumes of the whole thalamus and the anterior, lateral, ventral, intralaminar, medial, and posterior nuclei groups, using PLINK v2.0[18]. All GWAS accounted for age, age-orthogonalized age-squared, sex, scanning site, intracranial volume, and the first 20 genetic principal components to control for population stratification. The resulting $P$-values were Bonferroni-corrected for analyses of seven volumes. In addition, the GWAS for the six thalamic nuclei groups was run, both with and without whole thalamus as an additional covariate. The main text presents results for thalamic nuclei groups when accounting for whole thalamus volume, whereas GWAS results for the nuclei volumes are provided in the Supplementary Information.

We identified genetic loci related to thalamic volumes using the FUMA platform v1.3.5[29]. The settings and results of the FUMA analyses can be found at https://fuma.ctglab.nl/browse (FUMA ID 135-141). For these analyses, we used the UKB release2b White British as the reference panel. Independent significant SNPs were identified by the genome-wide significant threshold ($P < 7e − 9$) and by their independency ($r^2 \leq 0.6$ within a 1 mb window). We defined independent significant SNPs with $r^2 < 0.1$ within a 1 mb window as lead SNPs and genomic loci were identified by merging lead SNPs closer than 250 kb. All SNPs in LD ($r^2 \geq 0.6$) with one of the independent significant SNPs in the genetic loci were defined as candidate SNPs. We used the minimum $r^2$ to determine the borders of the genomic risk loci.

**Functional annotation, gene-based association, gene-set, tissue, and cell specificity, and protein–protein interaction analyses**. We functionally annotated all candidate SNPs of thalamus volumes using FUMA[29], which is based on information from 18 biological repositories and tools. FUMA prioritizes the most likely causal SNPs and genes by employing positional, eQTL, and chromatin interaction mapping[29]. The platform annotates significantly associated SNPs with functional categories, combined CADD scores[22], RegulomeDB scores[58], and chromatin states[29]. A CADD score > 12.37 indicates a deleterious protein effect[22], whereas the RegulomeDB score suggests the regulatory functionality of SNPs based on eQTLs and chromatin marks. Chromatin states show the genomic region's accessibility for every 200 bp with 15 categorical states predicted by ChromHMM 1.00 based on five histone modification marks (H3K4me3, H3K4me1, H3K36me3, H3K27me3, and H3K9me3) for 127 epigenomes[59]. Lower scores reflect higher accessibility in the chromatin state and to a more open state. Roadmap suggests the following 15-core chromatin states: 1 = Active Transcription Start Site (TSS); 2 = Flanking Active TSS; 3 = Transcription at Gene 5′ and 3′; 4 = Strong Transcription; 5 = Weak Transcription; 6 = Genic Enhancers; 7 = Enhancers; 8 = Zinc Finger Genes and Repeats; 9 = Heterochromatic; 10 = Bivalent/Poised TSS; 11 = Flanking Bivalent/Poised TSS/Enh; 12 = Bivalent Enhancer; 13 = Repressed PolyComb; 14 = Weak Repressed PolyComb; and 15 = Quiescent/Low[60].

We performed GWGAS and gene-set analyses with MAGMA v1.07[30] in FUMA on the complete GWAS input data. We excluded the major histocompatibility complex region before running the MAGMA-based analyses. MAGMA runs multiple linear regression to obtain gene-based $P$-values and the Bonferroni-corrected significant threshold was $P = 0.05/18158$ genes/7 volumes = 3.9e − 7. We ran gene-set analyses with hypergeometric tests for curated gene sets and GO terms obtained from the MsigDB[61].

We used LDSC-SEGs to assess the tissue and cell-type specificity of the GWAS findings (https://github.com/bulik/ldsc/wiki/Cell-type-specific-analyses). Following the analyses of Finucane et al.[28], we classified 205 tissues and cell types into nine categories for visualization, i.e., "Other," "Musculoskeletal-connective," "Liver," "Endocrine," "Digestive," "CNS," "Cardiovascular," "Blood/Immune," and "Adipose." We used a Bonferroni-corrected significance threshold of $P = 0.05/205$ tissues and cell types/7 thalamic GWAS = 3.5e − 5.

We conducted protein–protein interaction analysis using STRING to explore the functional relationships between protein-encoded thalamus-linked genes and exported the network to Cytoscape 3.8.2. for further analyses. We assessed protein–protein interaction enrichment and node degrees of the resulting network.

**Analyses of genetic correlation and overlap between thalamus volumes, cortical volumes, and ten brain disorders**. We derived volumes of 180 cortical regions in each hemisphere based on a multimodal parcellation of the cerebral cortex[36]. We then ran GWAS for volumes of each region summed for the two hemispheres with total cortical volume as a covariate and followed the implementation performed for the thalamus volumes. Next, we assessed genetic correlation between thalamic nuclei groups and cortical regions using LD-score regression, and adjusted the $P$-values across all performed analyses (7 thalamic volumes × 180 cortical volumes) using FDR correction in R statistics[62].

Furthermore, we obtained GWAS summary statistics for ADHD[41], ASD, SZ, BD, and MD from the Psychiatric Genomics Consortium[42–46], for AD from the International Genomics of Alzheimer's Project[47], for MS from the International Multiple Sclerosis Genetics Consortium[48], for PD from the International Parkinson Disease Genomics Consortium and 23andMe[27,49], and for GEP and FEP from The International League Against Epilepsy Consortium on Complex Epilepsies[50]. We performed similar analysis of genetic correlations between volumes and disorders as described for the thalamocortical correlations. In addition, we employed conjunctional FDR statistics[51,53,54,63,64] to assess polygenic overlap between the seven thalamic volumes and the ten brain disorders. The conjunctional FDR analyses were run using in-house software, available at https://github.com/precimed/pleiofdr/, MATLAB 2017a, and Python 3.7.4.

A review and mathematical description of this method when applied to neurological and psychiatric disorders were recently published by Smeland et al.[54]. In brief, the conjunctional FDR is defined by the maximum of the two conditional FDR values for a specific SNP. The method calculates a posterior probability that an SNP is null for either trait or both at the same time, given that the $P$-values for both phenotypes are as small, or smaller, than the $P$-values for each trait individually. The empirical null distribution in GWASs is affected by global variance inflation and all $P$-values were therefore corrected for inflation using a genomic inflation control procedure. All analyses were performed after excluding SNPs in the major extended histocompatibility complex (hg19 location chromosome 6: 25119106–33854733) and 8p23.1 regions (hg19 location chromosome 8: 7242715–12483982) for all cases, and *MAPT* and *APOE* regions for PD and AD, respectively, as complex correlations in regions with intricate LD can bias the FDR estimation.

**Reporting summary**. Further information on research design is available in the Nature Research Reporting Summary linked to this article.

## Data availability

Data used in this study were obtained from the UK Biobank, from the Psychiatric Genomics Consortium, 23andMe, the International Genomics of Alzheimer's Project, the International Multiple Sclerosis Genetics Consortium, the International Parkinson Disease Genomics Consortium, and from The International League Against Epilepsy Consortium on Complex Epilepsies. The PD GWAS partly included data from the 23andMe Consortium and is available through 23andMe to qualified researchers under an agreement with 23andMe that protects the privacy of the 23andMe participants. Please visit research.23andme.com/collaborate/#publication for more information and to apply to access the data. GWAS results are also available on the FUMA website; ID 135-141. The summary statistics for thalamic volumes of the present study are publicly available on GitHub.

## Code availability

This study used openly available software and code, specifically Freesurfer, Plink, LD-score regression, LDSC-SEG, Cytoscape, and conjunctional FDR.

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

## Acknowledgements

We were funded by the South-Eastern Norway Regional Health Authority 2015-078 (T. E.); 2013-123, 2017-112, and 2019-108 (O.A.A.); 2014-097, 2015-073, and 2016-083 (L.T. W.); by the Research Council of Norway (276082 (T.K.); 262656, 248778, 223273, and 273291 (O.A.A.); 204966 (L.T.W.); EEA 2014-2021, # 6/2019 (O.A.A.); 249795 and 273345 (L.T.W.); 283798 SYNSCHIZ (O.A.A.); Stiftelsen Kristian Gerhard Jebsen, the European Research Council (ERCStG 802998 BRAINMINT (L.T.W.)); NVIDIA Corporation GPU Grant (T.K.); 248828 Centre for Digital Life Norway (O.A.A.); the Ebbe Frøland foundation (T.E.); a research grant from Mrs. Throne-Holst (T.E.); and H2020 (# 847776) (O.A.A.). This research has been conducted using the UK Biobank Resource (access code 27412, https://www.ukbiobank.ac.uk/) and we thank the research participants and employees of the UK Biobank. We also thank the ADHD, ASD, SCZ, BD, and MD Working Groups of the Psychiatric Genomics Consortium, the International Genomics of Alzheimer's Project, the International Multiple Sclerosis Genetics Consortium, International Parkinson Disease Genomics Consortium, The International League Against Epilepsy Consortium on Complex Epilepsies, and 23andMe, Inc. for granting us access to their GWAS summary statistics, and the many people who provided DNA samples for their studies. This work was performed on the TSD (Tjeneste for Sensitive Data) facilities, owned by the University of Oslo, operated and developed by the TSD service group at the University of Oslo, IT-Department (USIT), and on resources provided by UNINETT Sigma2–the National Infrastructure for High Performance Computing and Data Storage in Norway.

## Author contributions

T.E and T.K. conceived the study, performed quality control, and analyzed and interpreted the data. A.S., O.F., D.v.d.M., S.B., O.S., V.J.K., L.T.W., and O.A.A. contributed with expertise on processing pipelines, conceptual feedback on analyses scopes, and/or interpretation of results. T.E. and T.K. wrote the first draft of the paper, and all authors contributed to and approved the final manuscript.

## Competing interests

T.E. is a consultant to BrainWaveBank and received speaker's honoraria from Lundbeck and Janssen Cilag. O.A.A. is a consultant to BrainWaveBank and HealthLytix, and received speaker's honoraria from Lundbeck. The remaining authors declare no competing interests.
