## [Peer Review File · Nature Communications]

Reviewers' Comments:

Reviewer #1:

Remarks to the Author:

The authors performed a GWAS on volumes of the whole thalamus and its six nuclei using brain MRI data from 30,432 genotyped individuals from the UK Biobank. For GWAS replication, MRI and SNP data from an additional 5,173 individuals from the UK Biobank were used. 42 (41 novel) genetic loci and 392 genes associated with volumes of the thalamus and its nuclei were identified. 96% of the loci were replicated. Overlapping genetic architectures consistent with thalamocortical connectivity was found. Pleiotropy analyses between thalamic volumes and ten psychiatric and neurological disorders showed shared variants for all disorders.

COMMENTS

1. In the introduction the authors write that they performed this study 'To further identify the genetic architectures of the thalamus and its nuclei and to assess their relationships with the cerebral cortex and brain disorders'. However the main aim of this study is to identify genetic risk loci and candidate genes for thalamic volume. Therefore, in my opinion, this aim is too broadly formulated.
2. For the GWAS the authors analyzed 30,432 genotyped individuals from the UK Biobank while for the replication phase another 5,173 individuals from the UK Biobank were used. It is unclear to me how the UK biobank imaging cohort was subdivided into this discovery and replication cohort. E.g. was this done by random sampling? Can the authors add information about this to the manuscript?
3. I find the circular Manhattan plots of whole thalamus volume, and of the volumes of the anterior, lateral, ventral, intralaminar, medial, and posterior nuclei separately difficult to read as there is much information in one figure. Could the separate Manhattan plots of these different analyses be added for clarity, for example in the supplementary info?
4. I cannot find any mentioning of genomic inflation factors for the different GWASs (GWAS whole thalamus volume, and GWAS of the volumes of the anterior, lateral, ventral, intralaminar, medial, and posterior nuclei separately). Could this information be added and discussed?
5. The authors describe the results of the replication phase as follows: 'We found that 96% of the lead SNPs from the discovery GWAS had the same effect direction in the replication (sign test; $P = 8.6e-14$). Moreover, 58% of the discovery lead SNPs had uncorrected $P < 0.05$ in the replication (Supplementary Table 9).' To make it easier to read I suggest to mention the number of SNPs that had the same effect direction in the replication instead of the percentage and also mention the SNPs (or the loci) that did not replicate.
6. Fig. 2 shows the results of the GWAS identifying 126 unique genes associated with thalamic volumes. Larger font size and higher position for gene names indicate greater Z-score. However, the genes indicated by the smallest font size are hardly readable.
7. To get further understanding of the mechanisms underlying differences in thalamic volume It would be interesting to see results of additional in-depth-analyses, such as results from cell-type/tissue-specific chromatin-based annotation analysis using stratified LDSC. Furthermore protein-protein interaction analyses could be performed to explore potential functional relationships between proteins encoded by the 392 candidate genes found associated with volumes of the thalamus and its nuclei.

Reviewer #2:

Remarks to the Author:

This manuscript described a genetic association study of thalamic nuclear volumes in the UK Biobank sample, with downstream gene-level annotation and analyses of genetic correlations with regional cortical anatomy as well as psychiatry traits.

I think this work is excellent. It provides a thorough analysis of an open question of importance to basic and clinical neuroscience (genetic architecture of the thalamus), and nicely puts these thalamic results in the context of thalamocortical connectivity and genetic risks for psychopathology. It is very well written - the analytic flow and findings are crystal clear - making the final product highly informative and helpfully transparent.

I only have a few comments/questions for consideration:

- Could the authors provide some rationale for the specific numbers split they chose for discovery vs. replication in the UKB?

- Showing similar replicability in e.g. the ABCD would be of great value if possible (but this is not essential for publication in my view).

- line 100 "The most strongly associated GWAS gene for the thalamic nuclei was DCDC1." Fig 2 clarifies which nucleus, but I would add this to the text too.

- line 117 "Thus, we ran GWAS in the discovery sample for volumes for each of the 180 cortical regions defined recently³⁵ and examined genetic correlations with the six thalamic nuclei volumes.". For the regional cortical GWA, was total cortical volume used as a covariate (to make the analyses analogous to use of total thalamic volume for thalamus)? This would be important to ensure that genetic correlations are being examined between equivalent thalamic and cortical phenotypes ... I.e. relative regional size.

Reviewer #1

The authors performed a GWAS on volumes of the whole thalamus and its six nuclei using brain MRI data from 30,432 genotyped individuals from the UK Biobank. For GWAS replication, MRI and SNP data from an additional 5,173 individuals from the UK Biobank were used. 42 (41 novel) genetic loci and 392 genes associated with volumes of the thalamus and its nuclei were identified. 96% of the loci were replicated. Overlapping genetic architectures consistent with thalamocortical connectivity was found. Pleiotropy analyses between thalamic volumes and ten psychiatric and neurological disorders showed shared variants for all disorders.

We are very grateful for the reviewer's comments and constructive suggestions, which have clearly helped us improve the manuscript.

Comments:

1. In the introduction the authors write that they performed this study 'To further identify the genetic architectures of the thalamus and its nuclei and to assess their relationships with the cerebral cortex and brain disorders'. However the main aim of this study is to identify genetic risk loci and candidate genes for thalamic volume. Therefore, in my opinion, this aim is too broadly formulated.

We agree with the reviewer that the aim was too broadly formulated and have rephrased accordingly the introduction:

"To further identify genetic loci and candidate genes for volumes of the whole thalamus and its nuclei, we accessed..."

2. For the GWAS the authors analyzed 30,432 genotyped individuals from the UK Biobank while for the replication phase another 5,173 individuals from the UK Biobank were used. It is unclear to me how the UK biobank imaging cohort was subdivided into this discovery and replication cohort. E.g. was this done by random sampling? Can the authors add information about this to the manuscript?

We thank the reviewer for pointing out that it was unclear how the UK biobank imaging cohort was divided into the discovery and replication cohorts. We conducted the discovery cohort analyses ($n = 30,432$) early winter 2020. As stated in the Methods section of the first manuscript version (lines 398-400), the results from the discovery GWAS were also used in the analyses of genetic overlap with

brain disorders and the discovery sample therefore excluded individuals with CNS diagnoses. At the time of the discovery cohort analyses, we used all data available to us after excluding individuals with these diagnoses ($n = 30,432$). Additional UK biobank imaging data then became available to us in spring 2020, making replication in independent data possible. To increase power of the replication sample, we also added data from the individuals with CNS diagnoses, that we had excluded from the discovery analysis, to the replication sample, yielding a replication cohort of $n = 5,266$. To increase clarity and detail, we have updated the Methods section and now state:

“We included raw T1-weighted 3D brain magnetic MRI data from $n = 30,432$ genotyped white British from the UK Biobank in the GWAS discovery sample. The results from the discovery GWAS were also used in the analyses of genetic overlap with brain disorders and the discovery sample therefore excluded individuals with CNS diagnoses. Additional data became available to us after we had performed the main discovery analysis and we combined this data with the previously excluded data of individuals with CNS diagnoses in order to maximize the sample size for a replication sample. The replication sample therefore comprised MRI and SNP data from an additional 5,266 white British of the general population in the UK Biobank¹⁵ irrespective of diagnoses”.

3. I find the circular Manhattan plots of whole thalamus volume, and of the volumes of the anterior, lateral, ventral, intralaminar, medial, and posterior nuclei separately difficult to read as there is much information in one figure. Could the separate Manhattan plots of these different analyses be added for clarity, for example in the supplementary info?

We thank the reviewer for the comment. The separate Manhattan plots corresponding to Fig 1d are provided in the Supplementary Fig. 2 of the revised manuscript.

4. I cannot find any mentioning of genomic inflation factors for the different GWASs (GWAS whole thalamus volume, and GWAS of the volumes of the anterior, lateral, ventral, intralaminar, medial, and posterior nuclei separately). Could this information be added and discussed?

We thank the reviewer for pointing out that we did not report the genomic inflation factor for the different GWASs. We have now added this to a new Supplementary Table 2 in the revised manuscript and we have added the following to the main text:

“The genomic inflation factors for the seven GWAS are listed in Supplementary Table 2. The strongest genomic inflation was observed for the ventral nuclei ($\lambda=1.168$), while the anterior nuclei showed the weakest inflation ($\lambda=1.105$).”

5. The authors describe the results of the replication phase as follows: ‘We found that 96% of the lead SNPs from the discovery GWAS had the same effect direction in the replication (sign test; $P=8.6e-14$). Moreover, 58% of the discovery lead SNPs had uncorrected $P < 0.05$ in the replication (Supplementary Table 9).’ To make it easier to read I suggest to mention the number of SNPs that had the same effect direction in the replication instead of the percentage and also mention the SNPs (or the loci) that did not replicate.

We thank the reviewer for pointing this out. We now report number of SNPs that had the same effect direction, number of SNPs with $P < 0.05$ in the replication and also mention the SNPs that did not replicate:

“We found that 53 out of the 55 lead SNPs from the discovery GWAS had the same effect direction in the replication (sign test; $P = 8.6e-14$). Moreover, 32 of the discovery lead SNPs had uncorrected $P < 0.05$, whereas 23 had uncorrected $P > 0.05$, in the replication (Supplementary Table 9).”

6. Fig. 2 shows the results of the GWAS identifying 126 unique genes associated with thalamic volumes. Larger font size and higher position for gene names indicate greater Z-score. However, the genes indicated by the smallest font size are hardly readable.

We thank the reviewer for the comment and have included a new Fig. 2, with larger font size, where color indicates Z-score:

Fig. 2 | GWGAS identifies 127 unique genes associated with thalamic volumes. Nineteen genes were associated with whole thalamus, 4, 29, and 17 genes were associated with volumes of the anterior, lateral, and ventral nuclei, and 37, 11, and 21 genes were associated with intralaminar, medial, and posterior nuclei volumes, respectively. Lighter font color and higher position for gene names indicate greater Z-score. Underlined gene names designate genes that were significantly associated with more than one volume, whereas gene names not underlined indicate genes associated with only one volume. Additional results of the GWGAS are presented in Supplementary Table 12. GWGAS; genome-wide gene-based association analysis.

7. To get further understanding of the mechanisms underlying differences in thalamic volume It would be interesting to see results of additional in-depth-analyses, such as results from cell-type/tissue-specific chromatin-based annotation analysis using stratified LDSC. Furthermore protein-protein interaction analyses could be performed to explore potential functional relationships between proteins encoded by the 392 candidate genes found associated with volumes of the thalamus and its nuclei.

We thank the reviewer for the fruitful suggestion and have added additional analyses using LDSC-SEG, as shown below in the new Supplementary Fig. 5. Here, we followed the multiple tissue and cell type analyses conducted by Finucane and colleagues and classified the 205 tissues and cell types into nine categories for visualization (Finucane *et al. Nature Genetics* 2018).

Supplementary Fig. 5 | Tissue and cell type specificity of the thalamic GWAS findings. We used linkage disequilibrium score regression applied to specifically expressed genes (LDSC-SEG) to assess the tissue and cell type specificity of the GWAS findings. Following the analyses of Finucane *et al.* (*Nature Genetics* 2018), we classified 205 tissues and cell types into nine categories for visualization, i.e., “Other”, “Musculoskeletal connective”, “Liver”, “Endocrine”, “Digestive”, “CNS”, “Cardiovascular”, “Blood/Immune”, and “Adipose”. We found that volumes of the whole thalamus and intralaminar and medial nuclei groups were enriched for “CNS” tissues and cell types, yet none of these associations was significant at a Bonferroni-corrected significance threshold of $P = 3.5e-5$ ($0.05/205$ tissues and cell types/ 7 thalamic GWAS).

We also added the following to the main text:

“To assess the tissue and cell type specificity of the thalamic GWAS findings, we used linkage disequilibrium score regression applied to specifically expressed genes (LDSC-SEG)²⁸. We found that volumes of the whole thalamus and intralaminar and medial nuclei groups were enriched for CNS tissues and cell types, yet none of these associations remained significant after Bonferroni-correction (Supplementary Fig. 5).”

The following was added to the Methods section:

“We used linkage disequilibrium score regression applied to specifically expressed genes (LDSC-SEG) to assess the tissue and cell type specificity of the GWAS findings [<https://github.com/bulik/ldsc/wiki/Cell-type-specific-analyses>]. Following the analyses of Finucane et al.²⁸, we classified 205 tissues and cell types into nine categories for visualization, i.e., “Other”, “Musculoskeletal-connective”, “Liver”, “Endocrine”, “Digestive”, “CNS”, “Cardiovascular”, “Blood/Immune”, and “Adipose”. We used a Bonferroni-corrected significance threshold of $P = 0.05/205$ tissues and cell types/ 7 thalamic GWAS = $3.5e-5$.”

Following the recommendation of the reviewer, we also added protein-protein interaction analyses to explore the functional relationships between proteins encoded by the candidate genes. These analyses were run using STRING [string-db.org] and we exported the network to Cytoscape 3.8.2. for further analyses. We added the following to the main text:

“We then performed protein-protein interaction analyses to explore the functional relationships between proteins encoded by the 391 candidate genes and detected a network with significantly more interactions than expected by chance (protein-protein interaction enrichment: $P < 1e-16$; Supplementary Table 14 and Supplementary Fig. 8). The most central network nodes were EGFR, RHOA, KANSL1, and NFKB1.”

We added the following to the Methods section:

“We conducted protein-protein interaction analysis using STRING [string-db.org] to explore the functional relationships between proteins encoded by thalamus-linked genes and exported the network to Cytoscape 3.8.2. for further analyses. We assessed protein-protein interaction enrichment and node degrees of the resulting network.”

We added a new Supplementary Table 14 and a Supplementary Fig. 8 to the revised manuscript:

Supplementary Fig. 8 | Protein-protein interaction analysis. We conducted protein-protein interaction analysis using STRING [string-db.org] to explore the functional relationships between proteins encoded by the 391 thalamus-linked genes and exported the network to Cytoscape 3.8.2. for further analyses. We detected a network with significantly more interactions than expected by chance (protein-protein interaction enrichment: $P < 1e-16$). The most central nodes were EGFR and RHOA, followed by KANSL1, NFKB1, EFTUD2, CRHR1, and RNF123. The figure depicts the resulting network with node color, node size and font size reflecting the respective node degrees (degree $d > 10$ in yellow). The most central nodes were EGFR ($d=32$) and RHOA ($d=29$), followed by KANSL1 ($d=15$), NFKB1 ($d=15$), EFTUD2 ($d=14$), CRHR1 ($d=13$) and RNF123 ($d=13$).

Reviewer #2

This manuscript described a genetic association study of thalamic nuclear volumes in the UK Biobank sample, with downstream gene-level annotation and analyses of genetic correlations with regional cortical anatomy as well as psychiatry traits.

I think this work is excellent. It provides a thorough analysis of an open question of importance to basic and clinical neuroscience (genetic architecture of the thalamus), and nicely puts these thalamic results in the context of thalamocortical connectivity and genetic risks for psychopathology. It is very well written - the analytic flow and findings are crystal clear - making the final product highly informative and helpfully transparent.

We thank the reviewer, and we are very grateful for the positive and constructive comments that have helped us improve the manuscript further.

I only have a few comments/questions for consideration:

- Could the authors provide some rationale for the specific numbers split they chose for discovery vs. replication in the UKB?

We thank the reviewer for pointing out that it was unclear how the UK biobank imaging cohort was divided into the discovery and replication cohorts. We conducted the discovery cohort analyses ($n = 30,432$) early winter 2020. As stated in the Methods section of the first manuscript version (lines 398-400), the results from the discovery GWAS were also used in the analyses of genetic overlap with brain disorders and the discovery sample therefore excluded individuals with CNS diagnoses. At the time of the discovery cohort analyses, we used all data available to us after excluding individuals with these diagnoses ($n = 30,432$). Additional UK biobank imaging data then became available to us in spring 2020, making replication in independent data possible. To increase power of the replication sample, we also added data from the individuals with CNS diagnoses, that we had excluded from the discovery analysis, to the replication sample, yielding a replication cohort of $n = 5,266$. To increase clarity and detail, we have updated the Methods section and now state:

“We included raw T1-weighted 3D brain magnetic MRI data from $n = 30,432$ genotyped white British from the UK Biobank in the GWAS discovery sample. The results from the discovery GWAS were also used in the analyses of genetic overlap with brain disorders and the discovery sample therefore excluded individuals with CNS diagnoses. Additional data became available to us after we had performed the main discovery analysis and we combined this data with the previously excluded data of

individuals with CNS diagnoses in order to maximize the sample size for a replication sample. The replication sample therefore comprised MRI and SNP data from an additional 5,266 white British of the general population in the UK Biobank¹⁵ irrespective of diagnoses”.

- Showing similar replicability in e.g. the ABCD would be of great value if possible (but this is not essential for publication in my view).

We agree with the reviewer that additional replication beyond the one that we have already performed in the manuscript will be of great value yet we feel that this would be better suited in a dedicated study. The currently available data for replication e.g., from the ABCD, comes from samples with a different age range and includes a broader span of ethnicities. Therefore, a dedicated study would be needed to investigate the diverse set of additional aspects to be looked at in this data.

- line 100 “The most strongly associated GWAS gene for the thalamic nuclei was DCDC1.” Fig 2 clarifies which nucleus, but I would add this to the text too.

We thank the reviewer for pointing this out and have added this to the text:

“The most strongly associated GWAS gene for the thalamic nuclei was DCDC1, which was linked to lateral nuclei volume.”

- line 117 “Thus, we ran GWAS in the discovery sample for volumes for each of the 180 cortical regions defined recently³⁵ and examined genetic correlations with the six thalamic nuclei volumes.”. For the regional cortical GWA, was total cortical volume used as a covariate (to make the analyses analogous to use of total thalamic volume for thalamus)? This would be important to ensure that genetic correlations are being examined between equivalent thalamic and cortical phenotypes ... I.e. relative regional size.

We thank the reviewer for the important comment and have clarified in the Methods section that total cortical volume was used as a covariate in the regional cortical GWA:

“We then ran GWAS for volumes of each region summed for the two hemispheres with total cortical volume as a covariate and followed the implementation performed for the thalamus volumes.”

NORMENT

Norwegian Centre for
Mental Disorders Research

Other revisions

We corrected an error in the number of thalamic-linked genes, i.e., the total number of genes associated with thalamic volumes was 391 – and not 392 as reported in the first manuscript version – and the total number of genes identified by the GWGAS was 127 – and not 126 as reported in the first manuscript version.

Reviewers' Comments:

Reviewer #1:

Remarks to the Author:

All of my comments have been addressed very extensively and satisfactorily by the authors. I have no additional comments left.

Reviewer #2:

Remarks to the Author:

The authors have fully addressed the comments made in my review. I support publication of this article.

Reviewer #1:

All of my comments have been addressed very extensively and satisfactorily by the authors. I have no additional comments left.

We thank the reviewer, and are very grateful for the positive comments and all constructive suggestions, which have clearly improved the manuscript.

Reviewer #2:

The authors have fully addressed the comments made in my review. I support publication of this article.

We thank the reviewer for all the comments and constructive suggestions, which have clearly helped us improve the manuscript.